# Changes in Perceived Mental Load and Motor Performance during Practice-to-Learn and Practice-to-Maintain in Basketball

**DOI:** 10.3390/ijerph20054664

**Published:** 2023-03-06

**Authors:** Alejandro Gutiérrez-Capote, Iker Madinabeitia, Elisa Torre, Francisco Alarcón, Jesús Jiménez-Martínez, David Cárdenas

**Affiliations:** 1Department of Physical Education and Sport, Faculty of Sports Science, University of Granada, 18071 Granada, Spain; agcapote@ugr.es (A.G.-C.); etorre@ugr.es (E.T.); j.jimenez@ugr.es (J.J.-M.); dcardena@ugr.es (D.C.); 2Sport and Health University Research Institute (iMUDS), 18007 Granada, Spain; 3Department of General and Specific Didactics, Faculty of Education, University of Alicante, 03690 Alicante, Spain; f.alarcon@gcloud.ua.es

**Keywords:** mental load, restrictions, motor performance, motor practice, training, basketball, skill acquisition

## Abstract

Background: Attentional resource allocation during sports practice is associated with the players’ perceived mental load. However, few ecological studies address this problem by considering the players’ characteristics (e.g., practice experience, skill and cognition). Therefore, this study aimed to analyse the dose-response effect of two different types of practice, each with different learning objectives, on mental load and motor performance by using a linear mixed model analysis. Method: Forty-four university students (age 20.36 ± 3.13 years) participated in this study. Two sessions were conducted, one based on a standard rules 1 × 1 basketball situation (“practice to maintain”) and one with motor, temporal and spatial restrictions in 1 × 1 tasks (“practice to learn”). Results: “Practice to learn” produced a higher perceived mental load (NASA-TLX scale) and a worse performance than “practice to maintain”, but was moderated by experience and inhibition (*p* = 0.001). The same happens in the most demanding restriction (i.e., temporal, *p* < 0.0001). Conclusion: The results showed that increasing the difficulty of 1 × 1 situations through restrictions harmed the player’s performance and increased their perceived mental load. These effects were moderated by previous basketball experience and the player’s inhibition capacity, so the difficulty adjustment should be based on the athletes themselves.

## 1. Introduction

Evidence shows that motor skills are flexible and not fully automated. Automatic and controlled cognitive processes develop jointly [1]. The flexibility of the cognitive control system allows the human being an adaptive motor performance, one which depends to a great extent on the efficient allocation of attentional resources and the information processing capacities of the performer [2]. An increase in the demands of the task entails a rise in the recruitment of processing resources and, consequently, in the mental workload [3]. The concept of mental workload has been widely studied in the motor learning area and defined as a multidimensional concept that arises from the relationship between imposed task demands and corresponding sensory capacities (e.g., visual, proprioceptive), cognitive capacities (e.g., attention, working memory, cognitive control), and motor capacities (e.g., planning, motor coordination) [4,5]. Its examination has been more limited during motor learning, where various aspects of the practice were manipulated [3].

Research approaches on sports motor learning have usually relied on different historically emergent models, such as repetitive learning (RL), discovery-based learning (DBL), methodical series of exercise (MSE), methodical game series (MGS), variability of practice learning (VP), contextual interference learning, and differential learning (DL) (for a comprehensive review of the models see [6].

Globally, the most predominant approaches are the cognitive and ecological approaches [7]. Within the cognitive approach, behavioural studies have indicated that the learning process and the efficiency of motor skill acquisition are generally influenced by various practical conditions concerning the skill levels obtained and the complexity of the tasks for learning [8,9,10]. The difficulty of the skill-learning task is a critical factor affecting the learning procedure; it can be defined as the difficulty level involved in executing a motor task within actual spatial and temporal constraints [11].

The challenge point framework has been proposed to explain the information processing underlying motor learning. According to this framework, there are optimal practice conditions based on the participants’ skill level and the task’s complexity [8]. Hodges and Lohseb [12] approximated this theory to sports science, establishing principles of action for designing learning practices and stages to guarantee the athletes’ evolution. Based on the challenge point framework, the authors highlight the need for the practice to have a certain degree of uncertainty as a first principle. As in the cognitive load theory, learning occurs by raising the level of challenge, which increases the recruitment of attentional resources and working memory, generating a mental workload increase and, therefore, the task’s difficulty [13,14]. The second principle of this framework establishes the need for an “optimal” level of difficulty or challenge adjusted to the pre-existing abilities of the athlete. The nominal difficulty of a task reflects the constant amount of difficulty, regardless of who is performing it and under what conditions [8]. For the same practice conditions that imply a quantity of information to be processed (nominal difficulty), the real difficulty of a task varies depending on the cognitive resources available to each athlete. This metric is called the functional difficulty of the task [8]. Learning is linked to functional difficulty and would therefore occur through adapting the cognitive system to the task’s demands, increasing the complexity of its response system. The degree of uncertainty generated by a task must be in a range that slightly exceeds the individual’s ability to cope with it. If there is too little information available, the efficiency of the processing system deteriorates due to the lack of stimulus to generate an adaptive response [9,15]. At the same time, too much information leads the system to collapse. For efficient motor learning, it is necessary that the amount of information available during the task’s performance and the information processing abilities of the athlete match. This optimal task difficulty for motor learning is called the optimal challenge point [9]. According to this framework, skill learning is expected to be most accelerated if functional task difficulty is matched to the optimal challenge point by manipulating nominal task difficulty and practice conditions based on the athlete’s skill level.

Although the challenge point framework as a model for motor learning research has been recently widely cited, to the best of our knowledge, there are no studies applying this framework to sports. These motor learning studies have made it possible to verify the impact of the difficulty level of the mental workload experienced upon motor execution or motor performance [9,16]. For this, traditional analysis at the group level were used through classical inferential statistics, without considering the data extraction methods in order to examine individual variability and its impact on cognitive-motor functioning. Shuggi et al. [3] identified specific individual patterns of cognitive-motor responses (for the same nominal difficulty), reaffirming that functional difficulty depends on experience and performance conditions but also on additional elements, such as individual processing characteristics [8]. In addition, the analysis revealed that the mental workload level ran parallel to the functional difficulty of the task, reinforcing the idea that its measurement could help quantify the functional difficulty experienced by individuals. Our purpose here is to analyse, in a sports context, specifically in basketball, the moderating role of the players’ initial level on the mental load and their performance of the task. Basketball is a useful context for study about the mental workload, since the players perform a significant cognitive effort due to their constant decision-making, allocating attentional resources to different sources of stimuli to anticipate their opponents’ and teammates’ intentions in a complex and variable environment. Players are also forced to process the contextual information under time pressure together with the emotional consequences of their decisions.

In addition, achieving the optimal challenge point requires understanding and a consideration of the athlete’s available cognitive resources. Cognitive control plays a key role in automated skills and strategic and tactical decision-making [1], the main objectives of which are to manage information and uncertainty. The functions performed by cognitive control systems, usually called executive functions (EFs), include control of attention, updating and processing of information (working memory), flexible integration of information related to the current situation and inhibition of inappropriate responses, planning of behaviour, decision making and problem solving [17,18]. A recent meta-analysis examining the influence of sports experience reported that elite athletes showed superiority in cognitive control tasks [19]. Unlike novices, expert athletes exhibited greater reactive and proactive response inhibition capacity [20,21,22] and more remarkable attentional ability, resulting in a higher level of working memory [23]. There is also evidence that cognitive control contributes to action execution [24].

Taking into account that individual characteristics, such as information processing capacity, are essential in determining the level of mental workload, it can be expected that beyond the group differences that may be found in the analysis of the performers’ dynamics in a task, the specific cognitive-motor responses of individuals can also be identified [25].

On the other hand, conditions that increase the performer’s cognitive demands have been shown to lead to better retention/learning (e.g., [26]). The adjustment of contextual variables in the manipulation of the practical task is the most frequently applied method for increasing uncertainty [27,28]. There is scientific evidence that justifies this practical proposal by which the demands of the task from the mental point of view are directly proportional to the degree of uncertainty it generates. The greater the level of the task’s uncertainty, the greater the resources it demands and, therefore, the greater the requirements of attentional control and working memory load [29].

However, in this type of practice, which Hodges and Lohseb [12] have called “practice to learn” (PtL), improvement occurs in the long term. The immediate effect is an increase in the number of errors. As the practice’s functional difficulty increases, the performance initially decreases. On the contrary facts, performance rises when the task’s functional difficulty is low and the number of errors reduced. Consequently, the task does not stimulate the adaptive capacity, and learning does not occur in the long term, leading the cited authors to call it “practice to maintain” (PtM).

Therefore, this study aims to analyse the dose-response effect of two different types of practice, according to their learning objectives, on changes in mental workload and motor performance using a mixed linear model analysis, which has been observed to be more suitable for use in studies with repeated measures design because it takes into account any specific patterns at the individual level, including them as random factors12 when there are likely to be correlations across the conditions of an experiment [30]. We hypothesize that the tasks to learn, which entail restrictions, are more challenging than the practice tasks to maintain. Therefore, the participants should experience greater functional difficulty by requiring more information processing. We also believe that those participants with higher levels of EFs would benefit by perceiving less functional difficulty.

## 2. Materials and Methods

In this section, the method used in the study will be explained in detail. To do so, the sample, study design, procedure, and variables, along with the measuring instruments used, will be described. Finally, the statistical procedures used for data analysis will also be described.

### 2.1. Participants

Forty-four students from the Faculty of Sport Sciences were recruited via email to participate in this study. To participate they had to meet the following inclusion criteria: (1) no history of cardiovascular, neurological, psychiatric or mental illness; (2) no muscle injury in the previous three months; (3) no concussion in the last 30 days; (4) no medication during the study period. All participants who were enrolled were engaged in physical exercise or sports practice for a minimum of 5 h per week (e.g., strength training or participation in a federated sports practice). In order to avoid the relationship between the study variables being due to chance, some variables that have been reported in the literature to have a moderating effect on mental workload were controlled. For this reason, participants were asked to fulfil the following requirements before performing the experimental sessions: avoidance of alcohol consumption 24 h beforehand [31], avoidance of caffeine consumption during the previous 12 h [32], and avoidance of strenuous exercise 48 h before arriving at the laboratory [33]. They were also required to sleep for at least seven hours the night before [34] and not to ingest any food four hours before each experimental session. Participation in this study was part of a set of extra-credit voluntary formative activities for students as a means to raise their grade in the basketball subject. In these activities, students are always awarded with 0.5 extra points on their final grade for their participation, regardless of the type of activity they decide to take part in.

Participants were divided into four groups according to their practical basketball experience in federated competitions: high experience (HE), ten or more years of experience; medium experience (ME), between 5 and 9 years of experience; low experience (LE), some unregulated practical experience; and no experience (NE), no practical experience. Participants’ demographic characteristics are listed in Table 1.

### 2.2. Design

A 4-factor repeated-measures within-subjects experimental design was used to test the acute effect of applying task restrictions in a basketball session on the participants’ mental workload perception, depending on their level of expertise. For this purpose, two counterbalanced experimental sessions were carried out: one with 1 × 1 free play tasks, following standard basketball rules (PtM), and another with 1 × 1 task restrictions to increase mental workload (PtL). Prior to the experimental sessions, a first familiarisation session (S1), a second session to assess participants’ cognitive ability (S2), and a third session to assess their physical condition and sports ability (S3) were carried out.

### 2.3. Procedure

#### 2.3.1. Pre-Experimental Sessions

In S1, participants received a general briefing about the study. They were also given an orientation about all the procedures and tests that they would perform during each session. At this point, participants gave their written consent. Subsequently, they were familiarised with all cognitive tests and questionnaires until they fully understood them. In S2, participants completed a battery of computer-based cognitive tests using the Psychology Experiment Building Language (PEBL) software program (Version 2.1, http://pebl.sourceforge.net (accessed on 2 February 2020) [35]. Each session lasted approximately 50 min, with a 3-min break between tests. In order to equalize cognitive assessments across all participants, the sessions were conducted between 10:00–12:00 h to control for circadian rhythms [36]. The sessions were conducted in a dedicated classroom with sufficient space between computers to avoid distractions between participants. The computers had Windows software, with a mouse enabled next to the participant’s dominant hand. Participants were instructed to sit comfortably about 60 cm from a 22′′ black-background computer screen. They were required to perform all the tests in the shortest possible time and as accurately as possible. The order in which the cognitive tests were performed was counterbalanced among all participants. In S3, participants from each experience group were evaluated to homogenize the level of opposition difficulty in the 1 × 1 situations through a cluster analysis (detailed below). As a result, pairings with a similar nominal difficulty level were determined.

Clustering analysis. The two-step cluster analysis was run to establish the pairs of 1 vs. 1 with a similar level. This statistic model is designed to efficiently handle large data sets in continuous and categorical variables and to determine the optimal number of clusters [37]. This clustering analysis uses a model-based distance measure that defines the distance between two clusters as the corresponding decrease in log-likelihood by combining them [38]. Firstly, the cases are sorted into pre-clusters and based on the distance measure; this non-supervised model decides whether a new cluster should be formed or if the case should be added to an existing cluster. The advantage of this step is that it reduces the size of the matrix, which contains the distance between all possible pairs of cases. Secondly, pre-clusters are pooled using a hierarchical clustering algorithm. Then the Bayesian information criterion (BIC) is used to select the best cluster solution. The first step of the cluster analysis included all the pre-experimental variables (detailed below in Section 2.4.1) which englobe physical fitness, sporting ability and cognitive capacity, since it has been observed to be a predictor of sports performance [39,40,41]. Their cluster quality output (i.e., distance measure analysis) was compared with each other to choose which variables were essential to determine the opponent’s difficulty for the subsequent analyses. The results showed that those variables were height, body mass, performance index rating (PIR; a valid method to evaluate a player’s overall performance. More details in Physical Fitness and Sporting Ability section), agility, skill and shoot performance. Lastly, the second step of the cluster analysis was performed, including only these variables, and the participants assigned in the clusters generated were randomly distributed in pairs of 1 vs. 1. An ANOVA test showed that the basketball level was similar, beginning with the experiment.

#### 2.3.2. Experimental Sessions

The session began with the distribution to each participant of the heart rate monitors and their setup (detailed below in Physiological Demands section). At this point, participants were told which experimental condition they would perform that day, they were reminded of the procedure to be followed, and any questions were answered. The experimental sessions consisted of a standardised 15-min warm-up consisting of jogging tasks at moderate intensity, dynamic stretching and progressive speed running [42]. Following the warm-up, players performed three 15-min practice blocks. Each block was divided into two similar 1 × 1 half-court basketball tasks lasting seven minutes. Between tasks, there was a one-minute break during which participants were not allowed to hydrate. At the end of each block, participants responded to the NASA TLX test (detailed below in Subjective Workload section). To avoid reciprocal influence between the responses of the two partners, they were separately interviewed by two researchers. Once they had answered the test, a 5-min break was left until the start of the next block, during which participants were allowed to hydrate. Finally, at the end of block 3 they sat on the floor for 15 min; the first 10 min were used for cooling and stretching [43], and the final 5 min were used to collect the heart rate monitors.

In the PtM, the tasks for each block were similar, consisting of free-play tasks following standard basketball rules. In the PtL, the order of task restrictions between blocks was counterbalanced across all pairs. Those restrictions were: (A) Motor restriction: The ball player only had three bounces per possession, (B) Temporal restriction: The ball player only had 5 s of possession, and (C) Space restriction: The playing space was reduced to only the centre part of the half-court (14 × 4.9 m). For each pair, all sessions were scheduled at the same daily time [36] and completed within a minimum of 72 h between lab visits. In all sessions, the same standard basketball of size 7 for boys and size 6 for girls (TF-1000; Spalding; Louisville, KY, USA) was used; in the case of a mixed 1 × 1, with a boy and a girl involved, one researcher was in charge of changing the ball in each attack phase.

### 2.4. Variables and Instruments

#### 2.4.1. Pre-Experimental Variables (S2–S3)

##### Cognitive Capacity (S2)

A set of tasks to measure the three core EFs, according to Diamond [44] (i.e., working memory, inhibition and cognitive flexibility) was selected to evaluate the possible moderating role of the participants’ cognitive capacity. Additionally, it has been proposed that inhibition-related processes are a family of functions rather than a single unitary construct [45,46]. According to Xie et al. [47] three types of inhibitions were measured: interference inhibition, rule inhibition, and response inhibition.

Go–No-go task: This task measured response inhibition [48]. Participants were presented with four quadrants on the screen. They had to make a motor response (right-click the computer mouse) to the presentation of a target letter. A single letter (“P” or “R”) was then presented in one of the quadrants for 500 ms with an inter-stimulus interval of 1500 ms. The test had two parts. In the first part, they had to right-click when they saw the letter “P” and not the letter “R”. In the second part, the rule was changed, and they had to right-click when they saw the letter “R” and not the letter “P”. Each part consisted of 10 practice trials and 160 experimental trials, divided into 128 target-letter trials (e.g., P-Go) and 32 non-target-letter trials (e.g., R–No-go). The main outcome variables were response accuracy, errors (both for an incorrect motor response and no response), response times, and the impact on accuracy and time from switching between the two parts of the test.

Number Stroop task: This measures cognitive/rule inhibition [49]. Participants were asked to count the (white) characters on the computer screen (with a maximum of three characters per trial). Once counted, they had to respond by pressing a number on the keyboard equal to the number of characters that had appeared on the screen (1-2-3). Each experimental trial began with a 1000 ms presentation of a white fixation “cross” symbol on the screen background. A stimulus was then presented for 2000 ms. The information could appear on the screen in three different ways: congruent (when the number of marks and the value of the number matched, e.g., 333), incongruent (when the number of characters and the meaning differed, e.g., 222) or neutral (the number of characters appeared with letters, e.g., HH). The test consisted of a block of 168 trials divided into 56 congruent, 56 incongruent, and 56 neutral trials presented in random order. The primary outcome variables were accuracy and mean response times per trial type.

Flanker task: This task was performed to assess interference control [50,51]. Participants were instructed to respond to the direction of a white arrow presented in the centre of the computer screen background. To do so, they had to press a button with their left index finger (left shift) when the arrow was pointing to the left (i.e., “<“) and press a button with their right index finger (right shift) when the arrow was pointing to the right (i.e., “>“). The test was divided into four flanking conditions: congruent (arrows oriented in the same direction, i.e., “< < < < < < < < <“ or “> > > > > > > >“), incongruent (arrows oriented in different directions, i.e., “> > > < < > > > >“ or “< < < > < < < <“), neutral (the central arrow is presented alone, i.e., “<“ or “>“), and dash (the central arrow has no distractor stimulus, i.e., “--<--” or “-->--”). The test consisted of a block of 172 trials divided into 12 practice trials at the beginning, followed by 40 congruent trials, 40 incongruent trials, 40 neutral trials, and 40 dash trials presented in a random order in each block. Each experimental trial began with a 500 ms presentation of a white fixation “cross” symbol in the computer screen’s background. Subsequently, a stimulus was presented for 800 ms, followed by an inter-stimulus interval of 1000 ms. The main outcome variables were accuracy and mean response times for each trial type.

N-back task: This task measured working memory [52]. Participants were instructed to use the index finger of their dominant hand to press on the computer keyboard (right shift button) when marking a response. Subsequently, different stimuli (white letters of the English alphabet 1 cm wide by 1.5 cm high) were presented. These appeared centred on the screen’s background for 500 ms with an inter-stimulus interval of 3000 ms. The test had two difficulties (1-back and 2-back), each with ten practice trials and 40 experimental trials. The first difficulty, called 1-back, consisted of pressing the computer keyboard each time the letter that appeared on the screen at that moment coincided with the one immediately preceding it. The second challenge, called 2-back, consisted of pressing the computer keyboard each time the letter displayed on the screen matched the previous two letters. The primary outcome variables were accuracy and response times.

Trail Making Test: This task measured cognitive flexibility [53]. Participants were asked to click the right mouse button as quickly and accurately as possible to connect blue circles arranged in a random sequence in ascending order on a black background. Before starting, participants were again informed that once the test type appeared on the screen, they would have time to examine it. The timing did not start until the first circle was clicked, which would be marked “1” regardless of the test. To ensure that the marked answer was on the correct circle, the circle would change colour to bold, and a line would be drawn from the previous circle to the selected one. If the wrong circle were clicked, it would not change colour, and no line would be drawn. The test was divided into two parts: In part A, the circles were numbered from 1 to 26 and had to be clicked on in numerical order (1-2-3-4). In part B, the circles appeared with numbers (1 to 13) and letters (A to M), and they had to click on them in alternating order (1-A-2-B-3-C). Four blocks were set for each test (A and B), and the main outcome variable was the time to complete the test.

Switcher task: This task compared the flexible switching between decision rules [54]. Ten targets were displayed on the computer screen, divided into five shapes and five colours. The targets were also marked with one of five letters. Each stimulus matched another in only one feature (colour, shape or letter). At the start of the test, a target was circled. At the top of the screen, the participant was asked to select the corresponding object based on the displayed feature of shape, colour or letter. To mark an answer, they were asked to respond as quickly and accurately as possible by clicking the right mouse button on the computer. As soon as the participant correctly matched the target, a different feature was specified at the top of the screen. They had to “switch” and select the new matching object based on that feature. The test consisted of nine blocks of 12 trials with different configurations preceded by ten practice trials. In the first three blocks, the configuration changed the targets repeatedly in two of the three features. In the subsequent three blocks’ configuration, the targets switched between the three features in a consistent order that varied in each configuration. In the last three blocks’ configuration, the targets changed randomly after each response, so neither the target nor the features could be anticipated before responding. The primary outcome variables were completion time and error rate (minimum number of clicks required to complete each block divided by the number of clicks made).

##### Physical Fitness and Sporting Ability (S3)

Height and body mass: Height and weight were measured with a measuring rod and a SECA 799 digital scale (Seca, Germany) with an accuracy of 0.1 kg. This made it possible to establish the body mass index (BMI; kg·m^−2^) from the measurements obtained.

Shooting: An adaptation of the tests used to measure shooting accuracy in conditions similar to real game situations, as described by Pojskic et al. [55] in their study, was performed. Participants made three-point shots with a ball from five outside positions, intending to make two consecutive baskets from all five positions in less than two minutes. The participant had to move halfway down the court between shots to return to the corresponding position. The number of positions from which the goal of making two consecutive baskets was achieved and the overall percentage of shots over the two minutes were counted. During the two minutes, another participant rebounded and passed the ball to the shooter.

Agility: Participants performed the agility T-test using the standard protocol proposed by Sporis et al. [56]. During the execution of the test, participants were verbally encouraged and asked to give their maximum possible effort. The total time was taken as the measurement parameter and they were allowed two trials separated by 2 min.

Ability: Participants were tested in the 20 m linear sprint and change of direction tests following the protocols established by Scanlan et al. [42]. The timing was measured manually by stopwatch. The fastest total time of the two tests was taken as the result.

Basketball performance level: Participants in each experience group competed against others in a 1 × 1 match-up in three-minute games with actual game rules. The only rule modification was that when the ball left the court, or the attack scored, the next attack started from the top of the three-point line in front of the basket. The number of wins and losses in the competition and their PIR were considered to evaluate each player. This method is widely used in European basketball leagues and other studies [27] to evaluate a player’s overall performance. It is calculated by applying the following formula: (points + rebounds + assists + steals + blocks + fouls committed) − (missed field goals + missed free throws + turnovers + blocks received + fouls committed). For this study, the formula was adapted by eliminating the variable “assists” as described by Camacho et al. [27] because, in 1 × 1 situations, this variable could not be evaluated.

#### 2.4.2. Experimental Variables (PtM-PtL)

##### Physiological Demands

Participants wore a Polar Pulsar RS800x (Polar Electro, Finland) to record their heart rates. These devices were used to monitor the participants’ levels in the intensity zone of interest. In this study, we aimed for participants to work at around 80–90%, as cognitive performance in higher-intensity exercise can be affected by physical and emotional fatigue [57]. For this purpose, we looked at the peak HR achieved during the session (%peak HR) calculated as a percentage of the maximum heart rate extracted using the 220-age formula [58]. Then, the Edwards training load (TL) was evaluated [59]. This method allows a coefficient to be established relative to five HR zones (50–60% peak HR = 1, 60–70% peak HR = 2, 70–80% peak HR = 3, 80–90% peak HR = 4, 90–100% peak HR = 5). This method has been widely used to calculate internal training load in basketball [60,61,62]. It correlates with the player’s external training load [63].

##### Subjective Workload

The NASA-TLX questionnaire [64] was used to assess the perceived subjective workload at the end of each session. This assessment was developed taking into account six dimensions: (1) physical activity, (2) mental activity (mental and perceptual effort required by the task), (3) temporal activity (pressure felt by the participant about the speed needed to respond to the task requirements), (4) performance/outcome (whether considered to have been successful during the task), (5) effort (the mental and physical difficulty with which the task was performed) and (6) frustration (negative feelings experienced during the task). In addition, an overall score can be obtained by averaging the six scores obtained in each dimension. Participants were asked to mark a scale corresponding to each dimension in this questionnaire. The scales go from lowest to highest, with a single exception for the scale corresponding to the dimension “performance/outcome”, which starts with “good” and ends with “bad”. The scale provides a minimum score of 0 points and a maximum of 100 points for each dimension.

##### Basketball Performance

The players’ PIR was recorded in both experimental conditions to determine the participants’ performance during the 1 × 1 situations. This made it possible to check: (1) whether the distribution of the participants into groups by practical experience was performed correctly; (2) whether the difficulty faced by the participants in the 1 × 1 situations was adjusted to each individual’s level; and (3) the impact of mental workload on performance. In order to achieve a stable level of motivation for the participants in both sets of experimental conditions, they were incentivized by the giving a gift of sports equipment to those who achieved a higher rating among the components of their experience group.

### 2.5. Statistical Analysis

Data summaries were computed for the whole sample. Firstly, a correlational analysis was carried out with the average value of the three blocks of each session to observe whether there was a relationship between performance (PIR) and mental load (NASA). Then, several linear mixed models (LMM) were performed to analyse: (1) differences in the PIR achieved in each block of the study; (2) the possible differences in PIR and mental load in both sessions; and (3) the possible moderator effect of the experience (i.e., HE, ME, LE, NE) and cognitive performance. Likewise, the same procedure was elaborated with the restriction scenario (i.e., 3-bounds, 5-s and limited space) with increased mental demands. LMM are an extension of linear models adding random effects in the linear predictor term to the regression setting. They allow us to model the dependence structure among dependent variables for longitudinal or repeated-measures data.

Experience groups and cognitive variables were added separately and combined into the session model (i.e., the model that contained the differences between the two sessions) to test whether experience and cognition moderated PIR and mental load. Then, the contribution of experience and cognition to model fit was tested by contrasting the models created against the session model. The same procedure was followed to test the possible moderation by the restrictions with more mental demands. This hierarchical test allowed for checking whether the increase in the proportion of explained variability attributable to the independent variable of interest (either experience or cognition) relative to a model without that independent variable compensates for the increase in complexity of the model, meaning that there exists a moderating effect. The Akaike information criterion (AIC) and the χ2 test were the indicators used to corroborate whether the new models fit the session model better, implying a moderation existed. The model with a lower AIC is considered to fit better.

LMM analyses were carried out with the lmer function from the nlme R package [65]. All quantitative predictors were also scaled and zero-centred before entering analyses. Effect sizes were calculated using the Nakagawa–Schielzeth approach [66].

## 3. Results

### 3.1. Descriptive and Correlational Analysis

Means and standard deviations for each variable of the study are displayed in Table 2. Regarding the correlational analysis between PIR and NASA of each block, the results showed that in the PtM there was a negative correlation between PIR and NASA-Overall (r = −0.371; *p* = 0.013), while in the PtL, PIR correlates negatively with NASA-Mental Activity (r = −0.334; *p* = 0.027), NASA-performance (r = −0.338; *p* = 0.025) and NASA-Overall (r = −0.572; *p* < 0.0001).

### 3.2. LMM—PIR Analysis between Blocks

The main outcome measure was the PIR obtained in the session, and each participant was included as a random factor (e.g., 1|participant). This model was compared with others in which the score of one block was subtracted. 

Regarding the PtM, the LMM showed that PIR block 1 had a significantly lower value in comparison with PIR blocks 2 and 3 [all PIR included model AIC (90.09); block 1 PIR subtracted model AIC (88.29; *p* = 0.047; R^2^ = 0.650)]. The same result is observed in the PtL [all PIR included model AIC (111.92); block 1 PIR subtracted model AIC (97.99; *p* < 0.0001; R^2^ = 0.563)]. This result is visually presented in Figure 1.

### 3.3. LMM—NASA and PIR Differences between Sessions and Moderation of Experience and Cognition

Firstly, to observe the differences between sessions, the main outcome was each dimension of NASA and PIR. Each participant was included as a random factor, and the models compared were the ones adding or not adding the session as a fixed factor. There were differences in Mental Activity [not including session in the model: AIC (253.36); including session in the model: AIC (208.68; *p* < 0.0001; R^2^ = 0.272)], Temporal Demands [not including session in the model: AIC (254.72); including session in the model: AIC (175.60; *p* < 0.0001; R^2^ = 0.567)], NASA-Overall [not including session in the model: AIC (254.67); including session in the model: AIC (207.48; *p* < 0.0001; R^2^ = 0.327)] and PIR [not including session in the model: AIC (200.43); including session in the model: AIC (189.25; *p* < 0.0001; R^2^ = 0.021)]. In each case, the analysis shows that there is a significant increment in the NASA dimensions and a significant decrease in PIR. 

Secondly, to see if Experience and Cognition moderated these dimensions in the PtL, new models were elaborated, including the four groups of Experience and Cognition, separately and combined. In this regard, the Reaction Time of Stroop [AIC = (125.60; *p* = 0.012; R^2^ = 0.135)] as well as Experience [AIC = (125.51; *p* = 0.015; R^2^ = 0.213) moderate Mental Activity [AIC = (129.85)], and including both variables (i.e., Experience and Stroop) seems to be the better model [AIC = (120.57; *p* = 0.001; R^2^ = 0.330)]. Finally, only Experience and not Cognition had a moderating effect on Temporal Demands [model without Experience: AIC = (129.85); model with Experience: AIC = (110.35; *p* < 0.0001; R^2^ = 0.445)], NASA-Overall [model without Experience: AIC = (129.85); model with Experience: AIC = (121.70; *p* = 0.002; R^2^ = 0.280)] and PIR [model without Experience: AIC = (129.85); model with Experience: AIC = (111.33; *p* < 0.0001; R^2^ = 0.433)]. The results of this epigraph are represented in Figure 2.

### 3.4. LMM—Checking the Restrictions with More NASA and PIR and Its Possible Moderation by Experience and Cognition

Each dimension of the NASA and PIR obtained in the three Restriction conditions (i.e., 3-bounds, 5-s and limited space) were considered separately as the main outcome. The models, and not the Restrictions as fixed factors, were compared. The analysis showed that there were only differences in Temporal Demands [model without Restrictions: AIC = (350.94); model with Restrictions: AIC = (345.98; *p* = 0.01; R^2^ = 0.033)], Effort [model without Restrictions: AIC = (361.28); model with Restrictions: AIC = (358.60; *p* = 0.03; R^2^ = 0.030)] and NASA-Overall [model without Restrictions: AIC = (375.61); model with Restrictions: AIC = (372.01; *p* = 0.02; R^2^ = 0.144)]. Checking each significant model, the Restriction producing a higher NASA value was the 5 s Restriction [Temporal demands (t = 3.01, *p* = 0.003); Effort (t = 1.93, *p* = 0.056); NASA-overall (t = 2.42, *p* = 0.030)]. 

To see whether Experience and Cognition played a moderating effect upon the Restrictions producing higher mental demands (i.e., 5 s), the steps followed were the same as described above: the models were compared, looking only to the NASA value, with other models including the four Experience groups and Cognition separately and combined. In the Temporal Demands score, the reaction time of Flanker [AIC = (127.57; *p* = 0.03; R^2^ = 0.09)] as well as Experience [AIC = (114.88; *p* < 0.0001; R^2^ = 0.384) had a moderate effect [model without fixed factors: AIC = (129.85)], and including both variables (i.e., Experience and Flanker) seems to be the better model [AIC = (109.86; *p* < 0.0001; R^2^ = 0.476)]. Finally, only Experience had a moderate effect in NASA-overall [model without Experience: AIC = (129.85); model with Experience: AIC = (114.79; *p* < 0.0001; R^2^ = 0.386)]. There was not any moderation effect from Effort. The results of this epigraph are represented in Figure 3.

## 4. Discussion

This study’s main objective was to analyse basketball players’ responses to different conditions of practice difficulty. First, we set out to test whether the NASA-TLX questionnaire would provide adequate quantitative measures of functional task difficulty. Secondly, it was expected to find that the difficulty introduced from the variability of the practice through the use of restrictions (motor, temporal and spatial) would increase each player’s mental load and decrease their performance. Moreover, finally, we expected that each player’s experience and cognitive abilities would moderate this effect.

The results of this study show that the PIR, a global indicator of motor performance in basketball, decreases with increasing task restrictions. This result confirms that we could adjust the functional task’s difficulty by manipulating the practice conditions’ variability. The challenge point framework predicts that success decreases as a function of an increase in nominal task difficulty during practice. Moreover, functional task difficulty is affected, on the one hand, by nominal task difficulty, and on the other, by the individual’s skill level upon encountering the task [8]. This study paired players with opponents of the same skill level. This procedure ensured that there would be no differences in skill level between the two sessions. Since the only difference between the conditions in the present study was nominal task difficulty, the variation in functional task difficulty between the two conditions is considered a consequence of nominal difficulty alone. Therefore, the functional difficulty of the task increased with the boost in the variability of the practice conditions through the use of the restrictions.

In addition, as has occurred in the field of motor learning, the results indicate that the NASA-TLX can serve as a helpful tool that quantitatively indicates the functional difficulty of the task in a real sports context. After correlating the average of the three NASA-TLX practice blocks, a relationship was found with the average PIR during both restriction sessions. Above all, these correlations were very high in the “practice to learn” session in the mental demand, time demand and performance dimensions (Table 2). Higher difficulty generated by the restrictions would increase the gap between the performance expected by the player and the actual one. The performance dimension of NASA-TLX gathers this perception. 

The task-related dimensions (mental demand, physical demand, and time demand) reflect task characteristics because they focus on the objective demands imposed by tasks. Furthermore, two of the task’s restrictions manipulated to increase the nominal difficulty showed the greatest relationship with these demands. It should be noted that the mental demand dimension of the NASA TLX has been considered essential to measure mental workload since it would reflect mental effort (e.g., [9]). On the other hand, no relationship was found with the dimension of physical demand. This outcome could be influenced by the intensity of the exercise, which is unlikely to affect motor performance.

The introduction of restrictions affecting the difficulty of the small-side game of 1v1 shifting it towards “practice to learn” resulted in a greater mental workload and consequently a worsening of the performance (PIR) in comparison with a session of “practice to maintain.” In motor learning, works such as those by Akizuki and Ohashi [9] and Shuggi et al. [3] reported similar results, finding that motor performance was altered by a too-high level of task functional difficulty. From the optimal challenge-point framework, an increase in nominal difficulty would have imposed a high level of challenge that would generally have overwhelmed most participants due to the increased demands of processing information relevant to performing the reaching task. This effort to process the information would have increased the mental demand, which, in our case, would be too high, resulting in lower performance in the 1v1 situation [9]. This increased mental demand can ultimately impair attentional focus [67,68] and physical performance [69]. It affects the ability of players to interact with environmental information [70] and, therefore, motor performance in the task, as has been verified in the sports field. Alder et al. [71] found in a laboratory study that the combination of physical and mental load aggravated the negative effect on the ability of soccer players to anticipate their actions. These results are also consistent with other laboratory studies in which the mental load experienced by participants increased as a consequence of time pressure [72,73]. On the other hand, the results agree with other current field studies that found a negative influence of mental workload on sports performance when applying task restrictions in their experimental sessions [27,74,75,76,77,78].

The increase in the mental load experienced by the players during the practice-to-learn task would reflect demands very close to the limits of the player’s current resources, showing less adaptive behaviour to the task, with deteriorated performance. This new environment would challenge them, producing beneficial practice conditions that could lead to better learning [26]. As established in the cognitive load theory, to learn, the practitioner must experience a mental load, which is the consequence of investing in a mental effort [14,79]. However, in the tasks given in “practice to learn”, the improvement occurs in the long term. The immediate effect is an increased number of mistakes [12]. As with the study players, performance initially decreases when the practice’s functional difficulty increases.

It should be noted that this decrease in performance and increase in mental load did not prevent the practice from continuing to be within an optimal challenge, as reflected by the performance improvements produced over time. It is also reflected in the fact that there was no increase in the frustration dimension of NASA-TLX. Both practice conditions, “practice to maintain” and “practice to learn”, fulfilled their function of stimulating and promoting the player’s adaptation. In both, performance increased throughout the session.

The present study also investigated two individual aspects of the players (sports experience and cognitive abilities) as potential moderators of the relationship between nominal task difficulty and functional difficulty, as well as sports performance. The fit results showed that the proposed mathematical model was the correct method to describe the relationship between difficulty and performance and difficulty and mental load. The fit between actual and modelled performance was statistically significant for the study players. The same thing happened with the mental load experienced. After increasing task difficulty, mental load changes are moderated by experience. The more experience the players have, the less the mental and temporary demands and the lighter overall mental load they experience. This same moderating effect is partially found in some cognitive abilities. Interference and response inhibition moderates the effect of difficulty on temporal and mental demands, respectively. The better the players’ cognitive performance, the less the mental load perceived when changes occur due to the task’s difficulty. A higher performance in interference inhibition decreases the perception of temporal demand, just as a greater response inhibition moderates the perception of mental demand, decreasing it. In the optimal challenge framework, it is established that functional difficulty, in addition to experience and practice conditions, also depends on individual processing characteristics [8]. Here, players with better interference (Flank) and response (Stroop) inhibition would have smaller functional difficulty resulting in smaller mental demands associated with better performance.

The best model that explains mental activity collects experience and response inhibition. Inhibitory control is related to sports performance and increases with greater athlete experience [22]. This pattern could be explained by the prevalence of proactive and reactive motor inhibition and its change according to the difficulty and player’s experience. Elite athletes are quicker to reach and maintain a constant level of proactive motor response inhibition [20], as well as exhibit behavioural and electrophysiological advantages (i.e., fewer cognitive resources required) by suppressing planned responses [80]. This assumption is in line with the extensive literature on elite sports, which has shown that highly-skilled athletes are better at strategically modulating their cognitive and motor resources according to the demands of specific tasks [81]. Open sports athletes deploy fewer attentional resources during uncertain situations by proactively updating the new rule of action [82]. Proactive control of the athlete could inhibit the incorrect response before the possible responses are activated [83], a more effective mechanism when active information is available in working memory. However, the cognitive control system, and therefore the use of a more proactive or reactive system, is mediated by an arbitration mechanism (or meta-control), which operates in a highly flexible way depending on the cognitive demand of the task [84]. Specifically, it is sensitive to the interaction between task difficulty variables and environmental uncertainty [85]. Thus, an increase in nominal difficulty could have initiated a shift from proactive to reactive at different times (for a review of this effect, see: [85,86]), according to previous experience, perceiving a lower mental activity in those with more cognitive resources and experience. 

This study presents novel results on the need to adjust the training tasks to the players’ skill level, since it has been shown that each group perceives changes in functional difficulty differently depending on their accumulated practical experience and initial cognitive resources. However, caution should be exercised, as more field studies are needed to test the impact of mental workload on sports performance when different restrictions on training tasks are applied. As suggested by Alarcón et al. [87], it seems logical to think that the degree of uncertainty imposed by a task should oscillate in a range that slightly exceeds the individual’s ability to cope with it. Therefore, during the players’ training process, the coaches must assess the individual skill level to adjust the nominal difficulty, but above all, the functional difficulty of the tasks [88].

### 4.1. Limitations

While our results have revealed the negative impact of nominal difficulty in a task-limited session, the current study has some limitations. Firstly, the sample size is small within each experience group, so a larger sample size and a larger number of studies that could replicate these results would be desirable. Secondly, only TL was controlled for as an external workload objective measurement, and it could have been enjoyable to incorporate other variables, such as cortisol and amylase, for the analysis of the internal objective workload.

Although there has been shown that an increment of the task’s functional difficulty results in an acute increase in mental workload and performance deterioration, the lack of testing the learning and transfer effects must be considered a limitation and corrected for future research.

### 4.2. Practical Applications

The results of the present study provide valuable information for coaches for the design of training tasks and workload adjustment. The analysis of the influence of manipulating the tasks’ restrictions allows for the creation of practice environments with a high levels of uncertainty in which players have to analyse and select the most relevant information presented to them in order to choose the best solutions [27]. In the constant search for individual player improvement by coaches, this study suggests different ways of manipulating the contextual variables of tasks to make players perceive a greater mental load. These findings are intended to guide coaches in the development of their sport activity planning on the need to adequately adjust the physical and mental load that stimulates short- and long-term adaptive processes [29]. To this end, it is proposed to pay special attention to the adjustment of the functional difficulty of the tasks [88], or in other words, to the actual resulting difficulty taking into account the athlete’s level of ability to cope with the demands of the task. Our results would support the need to generate training loads above a minimum threshold that would stimulate their adaptive capacity but below a maximum threshold that would cause the collapse of the system to respond to the imposed demands.

## 5. Conclusions

Increasing the difficulty of 1 × 1 situations through motor, temporal and spatial restrictions harmed players’ performance. This was accompanied by an increase in perceived mental workload, caused by a growth in the dimensions of mental activity and temporal demand. These effects were moderated by previous basketball experience and the player’s inhibition capacity, so the difficulty adjustment must be based on the athletes’ differences. On the other hand, the results of the present study support the use of NASA-TLX mental load to quantify functional difficulty in reduced game tasks in basketball, both for PtM and PtL. Our findings suggest that coaches can adjust practice conditions based on the mental demand, temporal demand, and performance dimensions of NASA-TLX scores. 

Consequently, coaches can make practice conditions more difficult by constraining the time pressure, playing space, or a player’s degree of freedom. The measurement of functional task difficulty tells coaches how to adjust the nominal task difficulty. Moreover, the value of these analytic dimensions informs coaches about the reach of the objectives of the practice design, enabling them to orientate more towards objectives of PtM, controlling the performance dimension (with a value around 53 in our results) or towards those of PtL, controlling the dimensions related to the task through the use of temporal restrictions, which has been the restriction with the highest difficulty values found (temporal demand around 74), but without exceeding the optimal challenge, so that the performance is improving over time.

## Figures and Tables

**Figure 1 ijerph-20-04664-f001:**
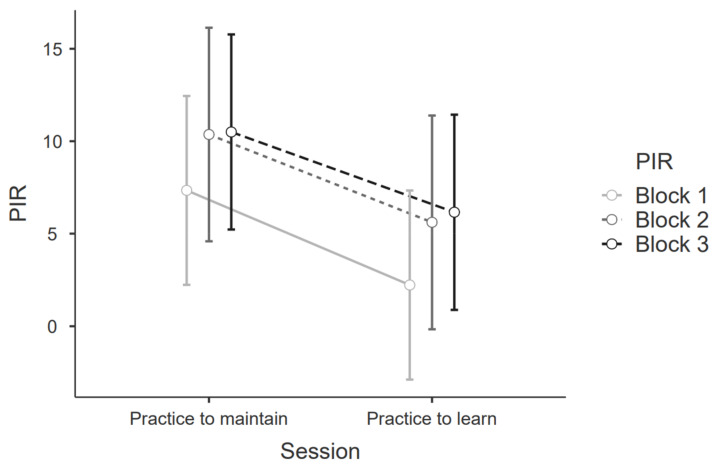
PIR per block obtained in each session.

**Figure 2 ijerph-20-04664-f002:**
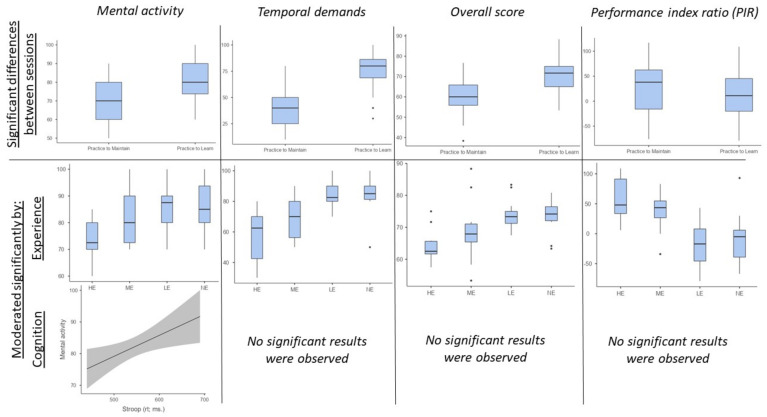
Differences between sessions and the possible moderation of experience and cognition. The only significant effect observed of cognition was Stroop on Mental Activity. HE: High experience; ME: Medium experience; LE: Low experience; NE: No experience.

**Figure 3 ijerph-20-04664-f003:**
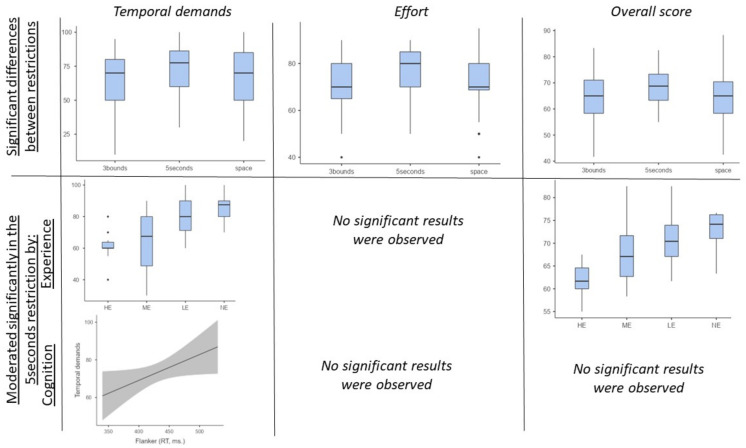
Analysis of which restriction was more demanding and its possible moderating effect upon experience and cognition. The only significant effect observed on cognition was Flanker on Temporal Demands. HE: High experience; ME: Medium experience; LE: Low experience; NE: No experience.

**Table 1 ijerph-20-04664-t001:** Participant demographics. Data are mean ± SD.

Variable	High Experience Group	Medium Experience Group	Low Experience Group	No Experience Group	All
N (M/F)	7/3	7/3	11/3	6/4	31/13
Age (yr)	21.10 ± 3.99	19.20 ± 1.69	20.64 ± 2.74	20.40 ± 3.89	20.36 ± 3.13
Height (m)	1.77 ± 0.08	1.80 ± 0.07	1.77 ± 0.06	1.70 ± 0.08	1.76 ± 0.08
Body mass (Kg)	74.25 ± 7.81	73.52 ± 9.08	68.96 ± 9.86	66.15 ± 7.04	70.56 ± 8.96
BMI (kg·m^−2^)	23.68 ± 1.85	22.58 ± 1.58	21.95 ± 2.31	22.80 ± 1.42	22.68 ± 1.90

Note: N: number of subjects; M: male; F: female; yr: years; m: meters; Kg: Kilograms; BMI: Body mass index.

**Table 2 ijerph-20-04664-t002:** Descriptives of NASA and basketball performance (PIR) between sessions and cognitive capacity.

Sessions					
Test	Variable	Practice to Maintain		Practice to Learn	
		Mean	SD	Mean	SD
NASA TLX	Mental Activity	69.7	9.85	82.0	10.75
	Physical Activity	72.4	8.73	74.9	8.66
	Temporary Demand	38.1	15.60	74.5	16.80
	Performance	53.9	17.18	57.5	14.96
	Effort	73.3	11.86	74.4	10.74
	Frustration Level	54.2	20.09	59.4	17.19
	Overall Score	60.2	7.41	70.5	7.52
PIR	Block 1	7.34	18.0	2.23	16.0
	Block 2	10.36	18.4	5.61	20.1
	Block 3	10.50	16.5	6.16	18.7
	Total session	28.25	48.4	14.41	48.5
**Cognitive Capacity**					
**Test**	**Variable**	**Mean**	**SD**		
Go–No-go	Accuracy	0.994	0.01		
	RT; ms.	445	49.4		
Stroop	Accuracy	0.956	0.03		
	RT; ms.	544	59.1		
Flanker	Accuracy	0.96	0.03		
	RT; ms.	430	37.9		
1back	Accuracy	0.95	0.07		
	RT; ms.	1600.40	259.26		
2back	Accuracy	0.91	0.1		
	RT; ms.	1717.82	254.05		
Ptrails	ms.	4287	5165		
Switcher	ms.	25,005	6034		

Note: RT: response time; ms: milliseconds.

## Data Availability

The corresponding author had full access to all the data in the study.

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
