# Peer review of "Changes in Perceived Mental Load and Motor Performance during Practice-to-Learn and Practice-to-Maintain in Basketball"

_ijerph, 2023, doi:10.3390/ijerph20054664_

Round 1
Reviewer 1 Report
Title: Changes in perceived mental load and motor performance during practice-to-learn and practice-to-maintain in basketball.
Overall: Although this study has found a relationship between manipulation of constraints and performance, I fail to see how this relates to learning as no testing of learning was performed. What is lacking is a test of learning i.e., the constraints group then performed without constraints after a break to see if transfer occurred. This is not mentioned as a limitation and/or a major requirement for future research. See below detailed comments.
Abstract:
Line 14: I suggest the authors review dynamical system theory and changes in constraints. As there is a number of important publications that have investigated constraints on learning.
Line 17: what is Mage? Remove the ‘M’. units go after the SD.
Line 20: what is ‘freedom’. Is this Degrees of Freedom? Is this the number of bounces? Need to make this clear in the methods section.
Line 25. Performance during the practice, yes. But how this relates to performance in a game or even over time is unknown.
Introduction:
Line 36. Remove ‘e.g.,’ – only need the reference.
Line 36. The following sentences are lacking in paragraph 1. What is meant by ‘increase in the demands of the task’, what has changed? Environment or organism perturbations? Or has the task become more complex due to changes in the goal of the task or rules that dictate the task goal? Would this be dependent on the task, environment, and organism relationship. Mental workload is an organism constraint which can be influenced by the task and the environment in which the task is being performed. Instead of ‘corresponding sensory’ etc. label this as organism constraints i.e. sensory, cognitive, and motor; then provide the examples.
Line 59: Is there research evidence to support this statement ‘too little information available, the efficiency…’?
Line 69: The authors do not mention any other frameworks, such as dynamical system theory (constraints led perspective), which has been used in exercise and sporting environments. As there are no comparisons to other frameworks, the authors have not clearly expressed why investigating this framework would be useful for motor learning in exercise and sport science. A stronger introduction of motor learning frameworks is needed.
Methods:
Line 129: How did the authors determine the participants to be ‘Healthy’? was a screening test performed? I do not see this to be relevant, just state the inclusion criteria.
Table 1. Weight is a force, measured in Newtons. Change this to Body mass (Kg).
Table 1. What is body max index? Do the authors mean body mass index? And what are the units?
Section 2.3.1. Was this a voluntary study? Or was it part of their studies? What happened to potential participants that were not recruited? As stated in section 2.1 participants were rewarded with 0.5 points on their final basketball grade. Would this create coercion?
Line 189: What variables? What variables were obtained from the cognitive tests? What did the cognitive test involve? Details are required or reference to the relevant section (i.e. see below).
Line 192: Change ‘weight’ to ‘body mass’.
Line 192: What is PIR? Abbreviation has been used without reference. Was agility part of the pre-experimental sessions? This clustering analysis section doesn’t seem to fit in the ‘pre-experimental sessions’. This would be better suited to the data analysis section.
Section 2.3.2
What is the make and model of heart rate monitor?
What type of warm-up?
What is the NASA TLX test? Reference needed.
Why did the participant need to sit for 15 mins at rest?
Section 2.4.1.2: Change ‘weight’ to ‘body mass’. BMI has units, please include them.
Agility: were both tests measured manually? With a stopwatch? What was the testers typical error?
Basketball performance level: What is the point guard position?(top of the three point line?) Some basketball offence strategies do not use named positions.
Results:
Discussion:
Line 488: what degree of freedom constraint was applied? There was spatial – area was decreased. Temporal – time to play; is DoF referring to the number of number of bounces? Why is this a DoF constraint? DoF is typically used in relationship to the body mechanics. Clarify on the use of DoF is needed throughout.
Line 493: The use of ‘variability’ doesn’t not work here. The authors manipulated constraints. Variability would be to change the constraints i.e., instead of 5 seconds in 1 trial it would be 8 seconds in the next etc.
Line 497: How were the participants paired with opponents of the same skill level? What skill? 1v1 skill? Agility? Sprint ability? Mental? Shooting? Or just experience? A major concern here is that a player can have a poor 1v1 ability but still be a very skilled basketball player. I think this is a major limitation that needs to be mentioned.
Author Response
Point-by-point replies
#Reviewer 1
Overall: Although this study has found a relationship between manipulation of constraints and performance, I fail to see how this relates to learning as no testing of learning was performed. What is lacking is a test of learning i.e., the constraints group then performed without constraints after a break to see if transfer occurred. This is not mentioned as a limitation and/or a major requirement for future research. See below detailed comments.
That is absolutely true, and the authors have included it as a significant limitation. We recognize that the title of the paper could lead to misunderstanding. However, the naming used in the paper as the study design is based on Hodges and Lohseb’s (2022) precedent trying to expand the challenge-point framework as a model of motor learning (Guadagnoli & Lee, 2004) and to apply it to sports coaching. According to the framework, increased difficulty during practice might be detrimental to performance in the short term but is ultimately beneficial for learning in the long term (Guadagnoli & Lee, 2004). Although we do not test the learning outcomes, the study results show a significantly different impact of two kinds of practice on acute performance and mental load, which could also lead to different effects on learning. Considering this limitation and calling to prudence, a paragraph explaining the importance of testing this hypothesis, including a learning test for future research, has also been added.
Hodges, N. J., & Lohse, K. R. (2022). An extended challenge-based framework for practice design in sports coaching. Journal of Sports Sciences, 40(7), 754-768.
Guadagnoli, M. A., & Lee, T. D. (2004). Challenge point: a framework for conceptualizing the effects of various practice conditions in motor learning. Journal of motor behavior, 36(2), 212-224.
Abstract:
- Line 14: I suggest the authors review dynamical system theory and changes in constraints. As there is a number of important publications that have investigated constraints on learning.
We realized that using the term “constraints” led the reviewer to consider contextualizing the study's goal on the dynamical system theory. To avoid the potential reader confounded mixing up different theoretical constructs, the word “constraint” has been replaced by “restriction”. One of the dependent variables, the mental workload, is not considered in the ecological research approach and is never analyzed since cognitive processes are not necessary to explain the perception-action coupling from this perspective. Although the authors acknowledge the relevance and enormous contribution of the complex dynamical system theory to sport pedagogy and sports practice, the particular focus of this study on the cognitive process made it impossible, or at least not adequate, to justify the research from this theoretical framework.
- Line 17: what is Mage? Remove the ‘M’. units go after the SD.
Thank you for the comment. The changes have been made (l. 18).
- Line 20: what is ‘freedom’. Is this Degrees of Freedom? Is this the number of bounces? Need to make this clear in the methods section.
The authors thank the reviewer for clarifying this point. According to Cardenas et al. (2015), uncertainty increases as the players' degree of freedom of action increases. In a task whose objective is associated with the learning or improvement of attacking skills, limiting the number of possibilities of action of the defenders reduces the difficulty; the same happens if the objective is defensive and is limited to the attackers. The type of limitation can be diverse and can vary from the establishment of rules prohibiting certain types of actions (no jumping to defend) to the use of motor conditioning (such as defending in basketball while bouncing a ball) (p. 94)". However, the authors understand that the information for readers may be incomplete by referring only to "freedom". To simplifying and get a clear and more direct message it has been changed by the term "motor restriction" (l. 20).
Cárdenas, D., Conde-González, J., & Perales, J. C. (2015). El papel de la carga mental en la planificación del entrenamiento deportivo. Revista de psicología del deporte, 24(1), 91-100.
- Line 25. Performance during the practice, yes. But how this relates to performance in a game or even over time is unknown.
We appreciate the reviewer’s statement since it offers an exciting standpoint for future research. Nevertheless, the article is focused on the effect of practice and not its relation to performance in competition, so it is unclear to us his requirement.
Introduction:
- Line 36. Remove ‘e.g.,’ – only need the reference.
Done (l. 36).
- Line 36. The following sentences are lacking in paragraph 1. What is meant by ‘increase in the demands of the task’, what has changed? Environment or organism perturbations? Or has the task become more complex due to changes in the goal of the task or rules that dictate the task goal? Would this be dependent on the task, environment, and organism relationship. Mental workload is an organism constraint which can be influenced by the task and the environment in which the task is being performed. Instead of ‘corresponding sensory’ etc. label this as organism constraints i.e. sensory, cognitive, and motor; then provide the examples.
We appreciate this comment since it made us realize the concept was unclear. To solve this issue, the term “demands of the task” has been changed to “task difficulty”.
- Line 59: Is there research evidence to support this statement ‘too little information available, the efficiency…’?
Two references supporting this statement have been included (l. 76).
- Line 69: The authors do not mention any other frameworks, such as dynamical system theory (constraints led perspective), which has been used in exercise and sporting environments. As there are no comparisons to other frameworks, the authors have not clearly expressed why investigating this framework would be useful for motor learning in exercise and sport science. A stronger introduction of motor learning frameworks is needed.
We appreciate the reviewer’s statement. We are absolutely agreed that there are other interesting and well-scientifically supported frameworks, such as dynamical system theory (constraints-led perspective). The authors considered that according to the experimental approach used for this research, which is based on specific previous evidence on motor learning areas (cited in the text), it could not make sense to spread out describing all the learning theories, which could be confusing for the readers. Nevertheless, according to the reviewer’s suggestion, not to mention this other approach could also be scarce, so two different things have been done. First, a paragraph has been added in the introduction section (ll. 44-48) summarizing the learning models historically developed in exercise and sport science. Finally, due to the term “constraint” connotations, clearly associated with dynamical system theory, this word was substituted by “restriction” to avoid misunderstanding.
Reference added:
Schöllhorn, W. I., Rizzi, N., Slapšinskaitė-Dackevičienė, A., & Leite, N. (2022). Always pay attention to which model of motor learning you are using. International journal of environmental research and public health, 19(2), 711.
Methods:
- Line 129: How did the authors determine the participants to be ‘Healthy’? was a screening test performed? I do not see this to be relevant, just state the inclusion criteria.
We appreciate the reviewer's contribution. We have rewritten the paragraph according to his comments and suggestions made by another reviewer (ll. 154-160). The paragraph is now much clearer.
- Table 1. Weight is a force, measured in Newtons. Change this to Body mass (Kg).
Changes have been made in Table 1. The terminology used is now more accurate and easily understood by the reader.
- Table 1. What is body max index? Do the authors mean body mass index? And what are the units?
Done. The units are described in the Table 1 next to their section. In addition, the abbreviation was added in the note below the table.
- Section 2.3.1. Was this a voluntary study? Or was it part of their studies? What happened to potential participants that were not recruited? As stated in section 2.1 participants were rewarded with 0.5 points on their final basketball grade. Would this create coercion?
The authors thank the reviewer for allowing us to clarify this section. The study was voluntary since students were offered an extra 0,5 points by choosing among several training activities. Therefore, they had the same opportunity to obtain the reward, although they chose a different academic task. In these activities, students are consistently awarded 0.5 points on their final grade for participation, regardless of the activity they decide to participate in. In addition, following the ethics committee statements, all students were previously informed that they were free to leave the research at any time they wished. As mentioned earlier in this section (2.1.), we have modified it accordingly. (ll. 167-171).
- Line 189: What variables? What variables were obtained from the cognitive tests? What did the cognitive test involve? Details are required or reference to the relevant section (i.e. see below).
Thanks to the reviewer for pointing out this issue because it helped us identify an error in the wording. It was written, "...The first step of the cluster analysis included all the variables explained above" when it should have been written "below". In addition, more detail has been provided in this new version, explaining why cognitive variables were included in the cluster analysis (ll. 221-223).
- Line 192: Change ‘weight’ to ‘body mass’.
Done (l. 226).
- Line 192: What is PIR? Abbreviation has been used without reference. Was agility part of the pre-experimental sessions? This clustering analysis section doesn’t seem to fit in the ‘pre-experimental sessions’. This would be better suited to the data analysis section.
According to the suggestion, the full name appears now before the abbreviation (ll. 226-228). Regarding agility (part of the pre-experimental session), we believe that the piece rewritten in point 13 of this reply will make this section of the manuscript easier to understand. Lastly, we acknowledge the reviewer's point concerning the location of the clustering analysis section. However, considering that this analysis aimed only to configure the experimental pairs carried out before beginning the experiment and there are no dependent variables, we believe that its original location is correct. Nevertheless, it will be done if the reviewer still considers that it should be moved to the data analysis section.
- What is the make and model of heart rate monitor?
The authors did not consider this section the ideal place to add this information. That is why the entire description of the heart rate monitors was developed in section "2.4.2.1. In any case, to make it easier for readers to find the information, the fragment: "detailed below in section 2.4.2.1." has been added after the mention of the heart rate monitor in the text indicated by the reviewer. (l. 235).
- What type of warm-up?
According to the reviewer's suggestion, this part has been rewritten in the text (now ll. 237-240.
Reference added:
Scanlan, A. T., Wen, N., Spiteri, T., Milanović, Z., Conte, D., Guy, J. H., ... & Dalbo, V. J. (2018). Dribble Deficit: A novel method to measure dribbling speed independent of sprinting speed in basketball players. Journal of sports sciences, 36(22), 2596-2602.
- What is the NASA TLX test? Reference needed.
After mentioning the NASA TLX test and aiming to facilitate the readers understanding, the text "detailed below in section 2.4.2.2." has been added. (ll. 243-244).
- Why did the participant need to sit for 15 mins at rest?
This part of the paragraph (ll. 247-249) has been rewritten and removed the word "rest" (which was probably used with its Spanish connotations in the field of sports training). This paragraph briefly explained that players were on the floor doing the cooling down and stretching part, which took 10 minutes. The reference on which this part of our methodology was based has also been added. The last 5 minutes were used to collect the heart rate monitors, avoiding the loss of any equipment and recording its data.
Reference added:
Contreras-Osorio, F., Guzmán-Guzmán, I. P., Cerda-Vega, E., Chirosa-Ríos, L., Ramírez-Campillo, R., & Campos-Jara, C. (2022). Effects of the Type of Sports Practice on the Executive Functions of Schoolchildren. International Journal of Environmental Research and Public Health, 19(7), 3886.
- Section 2.4.1.2: Change ‘weight’ to ‘body mass’. BMI has units, please include them.
Done. Changed in the document (ll. 353-355).
- Agility: were both tests measured manually? With a stopwatch? What was the testers typical error?
A trained rater measured both tests manually. Following Vicente-Rodríguez et al. (2011) manual measurements by a trained rater, using a stopwatch, seem valid to assess speed and agility fitness. To clarify that a stopwatch was used, the testo has been changed (l. 371).
Their study aimed to examine the interrater reliability (trained vs untrained raters) and criterion-related validity (manual vs automatic timing) of the 4 x 10-m shuttle run and 30-m running speed tests (times measured). The study comprised 85 adolescents (38 girls) aged 13.0-16.9 years from the Healthy Lifestyle in Europe by Nutrition in Adolescence study. The time required to complete the 4 3 10-m shuttle run and 30-m running tests was simultaneously measured (a) manually with a stopwatch by both trained and untrained raters (for interrater reliability analysis) and (b) by using photoelectric cells (for validity analysis). Systematic error, random error, and heteroscedasticity were studied with repeated-measured analysis of variance and Bland-Altman plots. In all cases, the systematic error for untrained vs trained raters and the untrained raters vs photoelectric cells was 0.1 seconds (p < 0.01). That is, untrained raters recorded higher times than trained raters. No systematic error was found between trained raters and photoelectric cells (p > 0.05). No heteroscedasticity was shown in any case (p > 0.05). The findings indicate that manual measurements by a trained rater using a stopwatch seem valid to assess speed and agility fitness.
Reference added:
Vicente-Rodríguez, G., Rey-López, J. P., Ruíz, J. R., Jiménez-Pavón, D., Bergman, P., Ciarapica, D., ... & HELENA Study Group. (2011). Interrater reliability and time measurement validity of speed–agility field tests in adolescents. The Journal of Strength & Conditioning Research, 25(7), 2059-2063.
- Basketball performance level: What is the point guard position? (top of the three point line?) Some basketball offence strategies do not use named positions.
We are grateful for the suggestion. Considering that this specific basketball terminology can confuse the readers, it has been changed by the following description: "top of the three-point line in front of the basket". (l. 375).
Results:
Discussion:
- Line 488: what degree of freedom constraint was applied? There was spatial – area was decreased. Temporal – time to play; is DoF referring to the number of number of bounces? Why is this a DoF constraint? DoF is typically used in relationship to the body mechanics. Clarify on the use of DoF is needed throughout.
The authors would like to thank the reviewer for thoroughly reading the paper. As mentioned in a previous commentary, the term "degrees of freedom" have been changed by “motor restriction” in the abstract (l. 20), methodology (l. 252) and discussion (l. 530) to avoid confusion.
In the called practice to learn (PtL), three types of restrictions were applied and counterbalanced across all pairs: (A) Motor restriction: The ball player only had three bounces per possession, (B) Temporal restriction: The ball player only had 5 seconds of possession, and (C) Space restriction: The playing space was reduced to only the centre part of the half-court (14 x 4.9 m).
- Line 493: The use of ‘variability’ doesn’t not work here. The authors manipulated constraints. Variability would be to change the constraints i.e., instead of 5 seconds in 1 trial it would be 8 seconds in the next etc.
The word “variability” has been deleted. The text is now as follows: “...Secondly, it was expected to find that the difficulty, introduced through the use of restrictions…” (ll. 529-530).
- Line 497: How were the participants paired with opponents of the same skill level? What skill? 1v1 skill? Agility? Sprint ability? Mental? Shooting? Or just experience? A major concern here is that a player can have a poor 1v1 ability but still be a very skilled basketball player. I think this is a major limitation that needs to be mentioned.
The cluster analysis was based on a group of indicators about the Physical Fitness and Sporting Ability level, described in section 2.4.1.2, including height, body mass, shooting, agility, ability and basketball performance level. The performance index rating (PIR) was used to evaluate the last. Its description is as follows: This method is widely used in European basketball leagues and other studies [22] to evaluate a player's overall performance. It is calculated by applying the following formula: (points + rebounds + assists + steals + blocks + fouls committed) - (missed field goals + missed free throws + turnovers + blocks received + fouls committed). For this study, the formula was adapted by eliminating the variable "assists" concerning the one described by Camacho et al. [22] because, in 1x1 situations, this variable could not be evaluated.
As the reviewer can check, the evaluation measures different basketball skills. However, the experimental control should ensure the players to face an opponent with a similar game level in the specific task conditions referred to 1x1 situations.
We agree with the reviewer that a player could have a poor 1v1 ability but still be a very skilled basketball player. Nevertheless, to guarantee a similar difficulty during the tasks, only 1x1 skills matter and must be similar between the players of each couple.

Reviewer 2 Report
This article is well written and well structured, and presents a very interesting study done in Spain that details the relationship between basically a player's mental state (the stress applied to the individual) and his physical performance, in this case in basketball.
The research carried out in this article seems to me to be in line with expectations, the references used are adequate and I believe that the objective of this study brings added value to research to this topic in particular.
The information in the tables was quite interesting and easy to understand, but concerning figures 2 and 3 I found them very confusing.
I feel that the conclusion should expose more of the amount of work that goes into this document (which is a lot), because as it stands it seems very vague and does not reflect at all what has been done in this investigation.
Overall, after some minor corrections, it is an interesting article.
Errors/questions/comments:
The pagination in the upper right corner seems to be badly formatted;
Line 73: "traditional analyzes" - "analysis" maybe?
Line 79: "traditional analyzes" - "analysis" maybe?
Line 82: "specifically in basketball" - Why basketball? There should be something here to justify this choice.
Line 119: "using a Mixed Lineal Model analysis which is more suitable because it takes into account any specific patterns at the individual level including them as a random factor" - Why is more suitable? Where is the reference or the justification?
Line 119: "Mixed Lineal Model" - "linear"?
Line 126: A short introduction here, before 2.1, introducing the subtopics would be nice. One paragraph is enough.
Line 128: "randomly recruited via email, participated in this study. All participants were healthy, physically active and engaged in physical exercise or sports practice for a minimum of 5 hours per week..." - This contradicts itself, so 44 random students were chosen, but by a simple coincidence they fit all the requirements listed below in the text? This paragraph needs to be revised.
Line 134: "For the duration of the study, they were asked to avoid alcohol consumption 24 hours before each session and caffeine consumption for 12 hours prior to participation" - Why? It seems obvious but there has to be a scientific justification.
Line 136: "They were also asked to avoid strenuous exercise 48 hours before arriving at the lab, to get at least seven hours of sleep the night before, and to eat a regular meal four hours before each session" - Once again, why? A justification is needed.
Line 205: "NASA TLX" - What is this? Where is the reference? (some pages below...)
Author Response
Point-by-point replies
#Reviewer 2
This article is well written and well structured, and presents a very interesting study done in Spain that details the relationship between basically a player's mental state (the stress applied to the individual) and his physical performance, in this case in basketball.
Thank you very much for your positive insight.
The research carried out in this article seems to me to be in line with expectations, the references used are adequate and I believe that the objective of this study brings added value to research to this topic in particular.
Once again, we appreciate the positive feedback.
The information in the tables was quite interesting and easy to understand, but concerning figures 2 and 3 I found them very confusing.
Figures 2 and 3 have been drawn again. The authors hope them to be easier to understand.
I feel that the conclusion should expose more of the amount of work that goes into this document (which is a lot), because as it stands it seems very vague and does not reflect at all what has been done in this investigation.
Following your suggestion, the English language has improved and some minor changes have been performed hoping that now this section is better than before.
Overall, after some minor corrections, it is an interesting article.
We appreciate your positive comments. We respond to each comment below. The parts that have undergone significant changes are highlighted in yellow in the new version of the manuscript.
Errors/questions/comments:
- The pagination in the upper right corner seems to be badly formatted;
The authors have checked the pagination of the manuscript and adjusted it again. We hope that no errors appear in this version.
- Line 73: "traditional analyzes" - "analysis" maybe?
We appreciate the thorough review of the paper. We have already made the changes to the document on the lines indicated (l. 88).
- Line 79: "traditional analyzes" - "analysis" maybe?
Changed in the document (l. 94).
- Line 82: "specifically in basketball" - Why basketball? There should be something here to justify this choice.
Basketball is a great context to study about the mental workload since the players do a great cognitive effort because of the constant decision making, allocating attentional resources to different sources of stimuli, to anticipate the opponents and teammates intentions in a complex and variable environment. Players are also forced to process the context information under time pressure and the emotional consequences of their decisions.
Following the reviewer’s suggestion, the previous paragraph has been introduced in the main text (ll. 98-103).
- Line 119: "using a Mixed Lineal Model analysis which is more suitable because it takes into account any specific patterns at the individual level including them as a random factor" - Why is more suitable? Where is the reference or the justification?
Thank you for the opportunity to clarify this issue. Recently, the mixed linear model has been dominating the psychological studies that use a repeated measures design and it might become the default approach to analyzing quantitative data. For instance, in studies like ours, the participants experience more than one of the manipulated experimental conditions (i.e. two experimental sessions, in which one of them there are 3 different conditions). The data-sets follow a hierarchical structure, and the mixed linear model allows this structure to be explicitly modeled. Part of this information has been included in the new version of the manuscript (ll. 140-142), and the following reference was included:
Meteyard, L., & Davies, R. A. (2020). Best practice guidance for linear mixed-effects models in psychological science. Journal of Memory and Language, 112, 104092.
- Line 119: "Mixed Lineal Model" - "linear"?
Changed in the document (l. 139).
- Line 126: A short introduction here, before 2.1, introducing the subtopics would be nice. One paragraph is enough.
Done. "In this section, the method used in the study will be explained in detail. To do so, the sample, study design, procedure, and variables along with the measuring instruments used will be described. Finally, the statistical procedures used for data analysis will be also described." (ll. 149-152).
- Line 128: "randomly recruited via email, participated in this study. All participants were healthy, physically active and engaged in physical exercise or sports practice for a minimum of 5 hours per week..." - This contradicts itself, so 44 random students were chosen, but by a simple coincidence they fit all the requirements listed below in the text? This paragraph needs to be revised.
This issue has also been highlighted for another reviewer who considered this section as doubtful and unclear with the reported information. Consequently, it has been rewritten to make easier its understanding (ll. 154-171).
- Line 134: "For the duration of the study, they were asked to avoid alcohol consumption 24 hours before each session and caffeine consumption for 12 hours prior to participation" - Why? It seems obvious but there has to be a scientific justification.
- Line 136: "They were also asked to avoid strenuous exercise 48 hours before arriving at the lab, to get at least seven hours of sleep the night before, and to eat a regular meal four hours before each session" - Once again, why? A justification is needed.
Response to comments 39 and 40:
Given the reviewers' doubts about this section, the authors have decided to rewrite it, paying particular attention to their comments. In order to conveniently substantiate the decision on the control of possible moderating variables of mental workload, the text is as follows: In order to avoid the relationship between the study variables being due to chance, some variables reported in the literature to have a moderating effect on mental workload were controlled. For this reason, participants were asked to fulfil the following requirements before performing the experimental sessions: avoidance of alcohol consumption 24 hours before (Rofey et al., 2007), caffeine consumption during the previous 12 hours (Papadelis et al., 2003), and strenuous exercise 48 hours before arriving at the laboratory (Costello et al., 2022). They were also required to fall asleep for at least seven hours the night before (Cesari et al., 2021) and not to ingest any food four hours before each experimental session.
References added:
Rofey, D. L., Corcoran, K. J., Tran, G. Q., Nabors, L. A., & Matthews, G. D. (2007). Demand on mental workload: Relation to cue reactivity and craving in women with disordered eating and problematic drinking. Addiction Research & Theory, 15(2), 189-203.
Papadelis, C., Kourtidou-Papadeli, C., Vlachogiannis, E., Skepastianos, P., Bamidis, P., Maglaveras, N., & Pappas, K. (2003). Effects of mental workload and caffeine on catecholamines and blood pressure compared to performance variations. Brain and cognition, 51(1), 143-154.
Costello, S. E., O’Neill, B. V., Howatson, G., van Someren, K., & Haskell-Ramsay, C. F. (2022). Detrimental effects on executive function and mood following consecutive days of repeated high-intensity sprint interval exercise in trained male sports players. Journal of sports sciences, 40(7), 783-796.
Cesari, V., Marinari, E., Laurino, M., Gemignani, A., & Menicucci, D. (2021). Attention-dependent physiological correlates in sleep-deprived young healthy humans. Behavioral Sciences, 11(2), 22.
- Line 205: "NASA TLX" - What is this? Where is the reference? (some pages below...)
This comment has already been addressed above for one of the other reviewers. Following their suggestions, "(detailed below in section 2.4.2.2)" has been added after the mention of the NASA TLX test (l.. 243-244).

Reviewer 3 Report
You can to find two auto cited references
Author Response
Point-by-point replies
- You can find two auto-cited references.
The authors would like to thank the reviewer for his comments. It is true that there are two self-citations in the manuscript, but both were necessary to focus the work. Moreover, these self-citations are part of previous work developed by the research group to which we belong, work that has allowed us to arrive at the hypotheses and objectives developed in this manuscript.
